# A Global Survey of Ethnic Indian Women Living with Polycystic Ovary Syndrome: Co-Morbidities, Concerns, Diagnosis Experiences, Quality of Life, and Use of Treatment Methods

**DOI:** 10.3390/ijerph192315850

**Published:** 2022-11-28

**Authors:** Vibhuti Samarth Rao, Stephanie Cowan, Mike Armour, Caroline A. Smith, Birinder S. Cheema, Lisa Moran, Siew Lim, Sabrina Gupta, Michael De Manincor, Vikram Sreedhar, Carolyn Ee

**Affiliations:** 1NICM Health Research Institute, Western Sydney University, Penrith 2751, Australia; 2Monash Centre for Health Research and Implementation, Monash University, Clayton 3168, Australia; 3Medical Research Institute of New Zealand (MRINZ), Wellington 6021, New Zealand; 4School of Health Sciences, Western Sydney University, Penrith 2571, Australia; 5Health Systems and Equity, Eastern Health Clinical School, Monash University, Boxhill 3128, Australia; 6School of Psychology and Public Health, La Trobe University, Melbourne 3086, Australia; 7School of Computer Science and Mathematics, Liverpool John Moores University, Liverpool L3 AF, UK

**Keywords:** polycystic ovary syndrome, PCOS, key concerns, diagnosis, satisfaction, quality of life, treatment methods, Indian women, survey, Indian migrant women

## Abstract

Background: Polycystic ovary syndrome (PCOS) is a common endocrinopathy that is highly prevalent in women of Indian ethnicity. Clinical practice guidelines do not adequately consider ethnic–cultural differences in the diagnosing and care of women with PCOS. This study aimed to understand co-morbidities, key concerns, quality of life (QoL), and diagnosis experiences of ethnic Indian women living with PCOS. Methods: Global online survey of ethnic Indian women of reproductive age living with PCOS. Results: Respondents (*n* = 4409) had a mean age of 26.8 (SD 5.5) years and reported having a family history of type 2 diabetes (43%) and PCOS (18%). Most of them (64%) were diagnosed with one or more co-morbidities (anxiety/depression being the most common). Irregular periods, cysts on the ovaries, and excess unwanted facial hair growth were their three top concerns. On average, women experienced symptoms of PCOS at the age of 19.0 (SD 5.0) and were diagnosed at the age of 20.8 years (SD 4.8). We report a one-year delay in seeking medical help and a seven-month diagnostic delay, which were associated with poor satisfaction with the information provided related to PCOS and its treatment options (*p* < 0.01). Women living outside India reported difficulty losing weight as their most key concern; however, they had higher dissatisfaction with the information provided on diet (OR, 0.74; 95% CI, 0.6 to 0.8; *p* = 0.002), exercise (OR, 0.74; 95% CI, 0.6 to 0.9; *p* = 0.002) and behavioural advice (OR, 0.74; 95% CI, 0.6 to 0.9; *p* = 0.004) than women living in India. Most women reported poor QoL in weight and emotion domains. Conclusions: Ethnic Indian women experience early onset of PCOS symptoms and delay in seeking professional help. Timely diagnosis, providing cultural-specific education related to lifestyle and weight management, and improving psycho-emotional support are key areas that should be addressed in clinical practice and future research.

## 1. Introduction

Polycystic ovary syndrome (PCOS) is a highly prevalent disorder characterized by oligo-ovulation and/or anovulation, hyperandrogenism, and polycystic ovarian morphology (PCOM) [1,2]. Women with PCOS experience lifelong metabolic (e.g., insulin resistance (IR), type 2 diabetes (T2D), adverse cardiovascular risk profiles) [3,4], reproductive (e.g., infertility, pregnancy complications) [5], and psychological consequences (e.g., anxiety, depression, poor quality of life (QoL), eating disorders) [6,7,8].

Experiences related to symptoms, diagnosis, and management of PCOS and its impact on QoL can differ markedly between women of various cultural backgrounds and ethnicities. Primarily studied in Caucasian women, emerging evidence has demonstrated that ethnicity and diagnostic criteria affect the phenotypic expression of PCOS [9,10,11,12]. For example, ethnic Indian women with PCOS present with earlier onset of PCOS symptoms and have more severe hirsutism, increased PCOM, metabolic risk (e.g., central obesity and insulin resistance), and reproductive symptoms (e.g., infertility and lower birth rates following in vitro fertilization) compared to Caucasian women [13,14,15,16].

Ethnic Indian women present with altered genetic susceptibility leading to early onset and increased severity of symptoms in most complex diseases, including PCOS [17]. Studies suggest that metabolic complications associated with PCOS, including central obesity, IR, T2D, and cardiovascular diseases are increasing rapidly among ethnic Indians irrespective of their geographic location [18]. For instance, according to a 20-year longitudinal follow-up study, Indian Asian migrants residing in the United Kingdom reported an incidence of T2D almost three times higher compared with the Caucasian controls [19]. With people of Indian ethnicity representing the world’s largest diaspora [20], we must consider PCOS phenotypes and comorbidities in ethnic Indian women living worldwide. Understanding the unique clinical manifestations of PCOS in ethnic Indians will help to better tailor diagnostic and management strategies to improve patient experiences.

PCOS is a complex condition and women have reported seeking help from multiple health professionals and having poor diagnosis experiences [21]. Diagnosis experiences associated with delay in diagnosis and inadequate information provision are known to influence treatment outcomes, impair QoL, and increase the risk of long-term consequences [22,23]. Dissatisfaction with the diagnosis experience and a lack of awareness and open dialogue about PCOS might be more intense in women with ethnically or culturally diverse backgrounds such as ethnic Indian women, considering the high prevalence of stigma associated with some of the PCOS manifestations such as menstruation, infertility, and obesity [6,24,25,26]. Suboptimal treatment experiences may also be heightened in ethnic Indian women who receive the same standards of care used for Caucasian women, which likely neglects the ethnic influence and cultural associations of PCOS [23,25].

Despite being a growing issue in ethnic Indian women [27,28,29], who present with worsened metabolic and reproductive phenotypes, we did not find any large global study that explored their concerns, PCOS symptomology, diagnosis experiences, and quality of life. We conducted a global survey of ethnic Indian women with the aim of (i) investigating co-morbidities, their key concerns and experiences with symptoms onset and diagnosis; (ii) exploring the provision of information and satisfaction with the information provided and overall diagnosis experiences; (iii) understanding the extent of different PCOS symptoms on their QoL.

## 2. Material and Methods

### 2.1. Ethical Approval and Consent

The research protocol was approved by the Western Sydney University Human Research Ethics Committee in February 2021 (reference H14103). A copy of the participant information sheet was provided electronically to participants before commencing the survey, which highlighted that the survey was anonymous, voluntary, and confidential and that participants could leave the survey at any time. Informed consent was implied upon the commencement of the survey.

### 2.2. Study Design and Research Instrument

The research team, consisting of content experts and researchers from various healthcare backgrounds (i.e., general practitioners, dietitians, women’s health researchers, public health practitioners, exercise physiologists, and yoga therapists), developed an online anonymous survey using Qualtrics software (Qualtrics Ltd., Provo, UT, USA). Upon development, the survey was piloted with ten adult ethnic Indian women—five residing in Australia and five residing in India. Their feedback was considered, and the final survey was amended accordingly.

The 72-item survey questionnaire included questions on the demographics, co-morbidities, family history of PCOS and T2D, diagnosis experiences (adapted from a previously published study) [21], important concerns of PCOS, health-related quality of life (HRQoL), health professionals consulted for PCOS, and treatment strategies used to manage the symptoms of PCOS, including Traditional, Complementary and Integrative Medicine (TCIM)) dietary and exercise interventions. We used a mixture of closed, multiple-choice, and open-ended questions.

The Modified Polycystic Ovary Syndrome Questionnaire (MPCOSQ) [30,31] was used to assess HRQoL. The MPCOSQ includes 30 questions from six HRQoL domains: emotional disturbance (8 items), weight concerns (5 items), infertility (4 items), acne (4 items), menstrual symptoms and predictability (4 items), and hirsutism (5 items). Each item was rated on a 7-point Likert scale where higher scores represent a better function. The domain scores are the sums of the scores for the items within each domain. A copy of the survey instrument can be found in Appendix A. The survey took approximately 20–30 min to complete. Features were enabled within Qualtrics that prevented multiple responses from either a single internet protocol (IP) address or the same computer.

### 2.3. Setting and Recruitment

The survey was open to any woman of ethnic Indian origin living in any country. Recruitment strategies included leveraging personal and professional connections of the research team, paid social media advertisements, and posting to various PCOS social media advocacy and support groups, Indian women’s groups, and global Indian migrant groups (including on Facebook, Twitter, and Instagram). Data collection occurred from mid-February to June 2021.

### 2.4. Eligibility

Eligibility criteria included women of self-reported Indian ancestry (ethnic Indian women either born in India or having at least one parent or grandparent who was born in India), age 18–55 years, with a self-reported diagnosis of PCOS by a medical doctor and who were able to read and understand English.

## 3. Data Analysis

Descriptive analysis was performed for normally distributed data (means and standard deviation), non-normally distributed data (median and interquartile ranges) and categorical variables (number and percentage). Body mass index (BMI) was calculated as (weight)kg/m^2^ (height). We categorized participants according to BMI as per guidelines for obesity and metabolic syndrome for Asian Indians, as normal (18.0–22.9 kg/m^2^), overweight (23.0–24.9 kg/m^2^), and obese (≥25 kg/m^2^) [32]. Missing values were only replaced for age, height, and weight data. Patterns of missing values were examined and median height (*n* = 411) and weight (*n* = 381) data related to the age groups was imputed [33]. For the MPCOSQ analysis, participants who completed the whole questionnaire were included, and a mean score for each domain was calculated.

Inferential statistics for between-group comparisons were performed using a chi-square test or independent t-test as appropriate. Correlations between variables were analysed using Spearman’s rank order and Pearson’s correlation as appropriate. Univariable and multivariable binary/ordinal regression analyses generated crude odds ratios (ORs) and 95% confidence intervals (CIs) for the relationship between the country of residence and the outcome variables (co-morbidities, key concerns, delay in diagnosis, delay in seeking help, information given/not given at the time of diagnosis, and satisfaction level with the information provided). A *p*-value of ≤0.05 was considered statistically significant. Analysis was performed in SPSS (version 28.0.1.0, 2021) [34].

## 4. Results

A total of 5546 women responded to our survey invitation. Of these, 733 were empty responses and were deleted, and 404 were not eligible to complete the survey (368 indicated that they had not been diagnosed with PCOS by a physician, while 36 indicated they were not of ethnic Indian ancestry). Therefore, a total of 4409 responses were included in the final analysis.

### 4.1. Sample Characteristics

The demographics of the participants are presented in Table 1. The mean age was 26.8 (SD 5.5) years, with more than two-thirds age 18–29 years (3189/4409, 72%). Body weight was 68.9 kg (SD 15.0) with a mean BMI of 26.8 kg/m^2^ (SD 5.6). Over 90% were born in India, with the remainder born in 33 different countries. Around three-quarters (2046/2780, 74%) of the participants were residing in India, while the top five countries of residence outside of India were Australia, the United States of America (USA), United Arab Emirates (UAE), Canada, and the United Kingdom (UK).

Almost all women (93%) were university educated, with 44% (1213/2780) having completed a postgraduate degree. The majority were in the workforce (1521/2806, 54%). Approximately 40% of the participants (1113/2780) were single, while 58% (1623/2780) were married or living in a relationship. More than two-thirds were nulliparous (2185/2780, 79%). Of the participants who reported a history of pregnancy, almost half (282/589, 48%) reported seeking medical help to conceive.

As per Table 1, 18% of women (719/3993) reported having a family history of PCOS. Among these, sisters (513/719, 66%) and mothers (250/719, 32%) were the most reported family members diagnosed with PCOS. Over one-third of women (1712/3963, 43%) had a family history of T2D. Of these, fathers (1290/1712, 75%) and mothers (828/1712, 48%) were the most reported family members with T2D.

### 4.2. Co-Morbidities Associated with PCOS

Most women (2555/4012, 64%) had been diagnosed with one or more co-morbidities, with psychological co-morbidities (61% diagnosed with anxiety, 36% with depression) and sleep disorders (27%) being the most common (Figure 1). Multiple co-morbidities were common, with women diagnosed with anxiety also being diagnosed with depression, chronic fatigue, thyroid disorders, sleep disorders, and eating disorders.

We found no significant association between the place of residency and the overall presence of co-morbidities (*p* = 0.118). However, there was a significant association between the place of residency and the presence of anxiety, T2D, and high cholesterol/triglyceride levels. The multivariable-adjusted model showed no statistically significant association between place of residency and anxiety, T2D, and high cholesterol/triglyceride levels.

### 4.3. Key Concerns of PCOS and Treatment Methods Used

Figure 2 represents the common signs/symptoms of PCOS that matter most to women living in India and outside India (*n* = 3550). Irregular cycles/periods (63%), difficulty losing weight (57%), and excess unwanted hair growth on the face (46%) were the most reported key concerns associated with PCOS. Other concerns included cysts on the ovaries (40%), increased tendency for weight gain (39%), excess hair loss (38%), and acne/pimples (32%). We found a significant association between the place of residence and most of the key concerns (except for depression, excess hair loss, and increased androgen levels). The multivariable-adjusted regression model showed that women living overseas were more likely to be concerned about excess unwanted facial hair growth (OR, 1.66; 95% CI, 1.1 to 2.3; *p* = 0.003) and difficulty losing weight (OR, 1.85; 95% CI, 1.2 to 2.7; *p* = 0.003), whereas acne (OR, 0.61; 95% CI, 0.4 to 0.9; *p* = 0.015) was their lowest concern compared than women living in India.

### 4.4. Experience with the Onset of PCOS Symptoms and Diagnosis

Table 2 describes the diagnostic data and symptoms at onset. On average, women experienced the first signs/symptoms of PCOS at age 19.0 (SD 5.0), first consulted a medical professional about their symptoms at age 20.0 years (SD 5.0), and were diagnosed at the age of 20.8 years (SD 4.8). Women waited for an average of 1.05 years (SD 2.3) from the first onset of symptoms before they attended their first medical consultation about PCOS, and there was a gap of 0.71 years (SD 2.3) between the first medical consultation and receiving a diagnosis. A Pearson’s correlation found a statistically significant, strong negative association between the age of the symptom’s onset and the year when medical help was first sought, *r* = −0.296, *p* < 0.001, and between the age of the symptom’s onset and the number of years taken to get a diagnosis of PCOS, *r* = −0.32, *p* < 0.001. A Spearman’s correlation was performed and found there was a statistically negative correlation between the number of doctors visited and both the year of onset of PCOS symptoms, *r*_s_ = −0.15, *p* < 0.001, and the year when medical help was first sought, *r*_s_ = −0.12, *p* < 0.001. Most participants were diagnosed by a gynecologist/obstetrician (3149/3839, 82%).

Irregular menstrual cycles/periods (3156/3824, 83%) were the most frequently reported first symptom, followed by cysts on the ovaries (in an ultrasound) (59%), excess unwanted hair growth on the face (48%), increased tendency for weight gain (48%), and difficulty losing weight (46%). We found a significant association between place of residence and cysts on the ovaries, excess unwanted hair growth on the face, difficulty losing weight, problems with ovulation, depression, infertility, acne/pimples, and excess hair loss. A multivariable adjusted regression model for first signs/symptoms onset revealed that women living overseas had lower odds of having depression (OR, 0.47; 95% CI, 0.2 to 0.8; *p* = 0.017) but higher odds of having excess unwanted facial hair growth (OR, 1.83; 95% CI, 1.2 to 2.5; *p* < 0.001) compared to women living in India.

### 4.5. Provision of Information and Satisfaction with the Information Provided at the Time of Diagnosis

According to Table 3, only 34% of women (1224/3595) were satisfied with the manner of diagnosis. Around one-third of women (34%) did not receive information on emotional support, long-term complications (35%), or behavioural advice to improve diet or exercise (30%). Of the women who did receive information from their healthcare providers regarding treatment and management options, more than two-thirds of the women reported dissatisfaction/indifference with the information provided on emotional support (75%) and treatment options (72%). Only one-third were satisfied with the information provided about PCOS (37%), long-term complications (34%), treatment options (36%), and dietary advice (22%).

#### 4.5.1. Provision of Information and Level of Satisfaction with the Country of Residence

We found a statistically significant association between some aspects of the provision of information (PCOS, long-term complications, and exercise) and country of residence (Table 4). The univariable analysis found that ethnic Indian women living outside India were more likely to report receiving information related to PCOS (OR, 1.30; 95% CI, 1.0 to 1.6; *p* = 0.039) or long-term complications (OR, 1.20; 95% CI, 1.0 to 1.4; *p* = 0.042); however, they were less likely to report receiving information about exercise (OR, 0.74; 95% CI, 0.6 to 0.9; *p* = 0.004) than women living in India. The odds of receiving information related to PCOS (OR, 1.40; 95% CI, 1.0 to 1.8; *p* = 0.014) and long-term complications (OR, 1.23; 95% CI, 1.0 to 1.4; *p* = 0.030) were higher in these women even after adjusting for other demographic co-variables. The multivariable-adjusted model showed that women living outside India were less likely to be satisfied with the information provided on diet (OR, 0.74; 95% CI, 0.6 to 0.8; *p* = 0.002), exercise (OR, 0.74; 95% CI, 0.6 to 0.9; *p* = 0.002), and behavioural advice (OR, 0.74; 95% CI, 0.6 to 0.9; *p* = 0.004).

#### 4.5.2. Provision of Information and Level of Satisfaction with Delay in Seeking Help

According to Table 4, we found a significant association between some aspects of diagnosis experiences (information not received for exercise and emotional support, *p* < 0.05) and delay in seeking help. Multivariable adjusted analysis showed significantly reduced odds of receiving information regarding exercise (OR, 0.77; 95% CI, 0.6 to 0.9; *p* = 0.011) and emotional support (OR, 0.83; 95% CI, 0.7 to 0.9; *p* = 0.037) associated with delay in seeking help.

The level of satisfaction with overall diagnosis experience and information received about all aspects of PCOS and its management (except for long-term information) were significantly associated with the delay in seeking help (*p* < 0.05). In a multivariable-adjusted analysis, we found reduced odds of satisfaction with the information provided for treatment options (OR, 0.71; 95% CI, 0.6 to 0.8; *p* < 0.001), diet (OR, 0.80; 95% CI, 0.6 to 0.9; *p* = 0.013), behavioural advice (OR, 0.69; 95% CI, 0.5 to 0.8; *p* < 0.001), and emotional support (OR, 0.62; 95% CI, 0.5 to 0.7; *p* < 0.001), with delay in seeking help.

#### 4.5.3. Provision of Information and Level of Satisfaction with Delay in Diagnosis

Information provision for PCOS, long-term complications, and treatment options were associated with delay in diagnosis (*p*< 0.05). Reduced odds of receiving information for PCOS (OR, 0.62; 95% CI, 0.4 to 0.8; *p* < 0.001) and long-term complications (OR, 0.81; 95% CI, 0.6 to 0.9; *p* = 0.031) were associated with delay in diagnosis in a multivariable-adjusted analysis (Table 4).

Significantly reduced odds of satisfaction with overall diagnosis (OR, 0.69; 95% CI, 0.5 to 0.8; *p* < 0.001), and the information given on all aspects of diagnosis; PCOS (OR, 0.68; 95% CI, 0.5 to 0.8; *p* < 0.001), long-term complications (OR, 0.70; 95% CI, 0.5 to 0.8; *p* < 0.001), treatment options (OR, 0.73; 95% CI, 0.6 to 0.8; *p* = 0.002), diet (OR, 0.74; 95% CI, 0.6 to 0.9; *p* = 0.004), exercise (OR, 0.68; 95% CI, 0.5 to 0.8; *p* < 0.001), behavioural advice (OR, 0.72; 95% CI, 0.5 to 0.8; *p* = 0.002) and emotional support after the diagnosis (OR, 0.78; 95% CI, 0.6 to 0.9; *p* = 0.034) were associated with delay in diagnosis.

### 4.6. Health Professionals and Treatment Methods

Table 5 describes the health professionals seen and different treatment approaches used by the participants to manage symptoms of PCOS. Most women (2812/3117, 90%) consulted one or two health professionals for their PCOS-related issues. Gynecologists/obstetricians were the most frequently visited (2809/3117, 90%) followed by general practitioners/family physicians (26%), and allied health professionals (20%).

More than half of women were using the combined oral contraceptive pill (1613/2547, 63%), followed by metformin (41%), and anti-androgen drugs (25%). The majority practised yoga (1775/3027, 59%), followed a specific diet (1978/2921, 68%), and were actively engaged in some kind of physical activity (2241/2921, 77%) to manage their symptoms. A large proportion of participants (1942/3130, 63%) reported using ingestible TCIM to manage PCOS, with Ayurveda (traditional Indian medicine) being the most used (1159/1942, 60%).

### 4.7. Health-Related Quality of Life Using MPCOSQ

Figure 3 presents a summary of the mean scores of the six domains of the MPCOSQ for the respondents (*n* = 2020). The lower the score, the greater the negative impact on HRQoL. The total mean score of all six domains was 3.2 (SD 1.0). We found that HRQoL was lowest in the domain of weight (2.7, SD 1.6), followed by emotions (3.0, SD 1.3) and menstrual cycle (3.1, SD 1.2). Acne (3.9, SD 1.7) and infertility (3.8, SD 2.0) were the highest-scoring HRQoL domains. An independent sample t-test was performed to compare QoL in women living in India and outside India. There was a significant difference in QoL in the domains of weight, *t*(1788) = 3.82, *p* <0.001; body hair, *t*(1788) = 3.73, *p* = 0.011; and infertility, *t*(1788) = 1.33, *p* = 0.019, between women living in India and outside India.

## 5. Discussion

In this large international survey, we found that most ethnic Indian women with PCOS present with multiple co-morbidities and frequently report symptom onset in their late teens. Overall, women reported a one-year delay in seeking medical help, and a seven-month diagnostic delay from symptom onset was associated with lower odds of receiving information and higher odds of dissatisfaction with the information provided related to PCOS and its management at the time of diagnosis. We observed that women living overseas were more likely to receive information related to PCOS than women living in India; however, they were less likely to be satisfied with the information given about diet, exercise, and behavioural advice. The key concerns that women had about PCOS were irregular cycles, difficulty losing weight, and unwanted hair growth on the face. We found significant mean differences in some domains of QoL: weight, body hair, and infertility depending on the country of residency. Women visited only one or two health professionals and used different treatment modalities, including TCIM.

Most women in our study experienced signs and symptoms of PCOS between the ages of 13–20 years and received a PCOS diagnosis between the ages of 17–24 years. These findings are consistent with previous research that indicates that South Asian women are experiencing symptoms including irregular cycle onset at an early age and hence are diagnosed at a younger age (between 16–25 years) than Caucasians (between 20–30 years) [23,35,36,37]. When PCOS symptoms present early in life (preadolescence and adolescence), women may be at a higher risk of developing complications [38] and experience a lower HRQoL [39], emphasizing an increased need for timely and appropriate diagnosis and adequate information provision and support following diagnosis. For example, alteration in the menstrual cycle more than two years post menarche during adolescence could be an important means for identifying adolescents at higher risk of developing PCOS and metabolic syndrome [40]. The new diagnostic criteria for diagnosing PCOS in adolescents can help clinicians identify women at risk for developing PCOS and plan better care provisions [41]. Therefore, it is important to carry out regular follow-ups with such women to explore the complexities involved in PCOS diagnosis and treatment.

While the time to diagnosis in our study (0.7 years, SD 2.3) was somewhat consistent with research in Caucasian women (which shows a wide variation of 43% ≤ six months and 34% ≥ two years) [22], we observed a considerable delay of one year (SD 2) in seeking treatment following initial symptom onset. This is longer than the three-month delay in seeking treatment reported by previous research from India (*n* = 275), where it was suggested that a key barrier to timely diagnosis was their propensity to endure symptoms silently until the severity was no longer tolerable [23]. Although the majority of participants were young, university educated, and working for wages, they delayed seeking help for their ongoing symptoms. Delayed treatment seeking was associated with lower odds of receiving emotional support at the time of diagnosis and dissatisfaction with the information provided. This suggests that socio-cultural factors need to be explored in greater detail while caring for these women.

There are many possible reasons for the delay in seeking help. PCOS is a women’s health issue that relates directly to menstruation/fertility, and it is possible that young women may be hesitant to discuss these issues, as they may believe that the symptoms will subside on their own over the course of time [23]. For more than 80% of women in our study, irregular menstruation was the first symptom of PCOS, which might suggest that women are trying different ways to manage their periods before seeking help. A previous study from India (*n* = 100) found that approximately half of women age 25–55 years were not comfortable talking about one or more women’s health issues due to the societal taboo associated with them, and 84% of women experienced discrimination and judgment related to their menstruation [42]. For women in our study, irregular periods were their top concern related to PCOS (63%). This is similar to a previous study conducted by Jain et al., where they reported irregular periods (66%) as the third most common symptom of PCOS in Indian women [43]. Therefore, educating women about menstruation and its irregularities is an important aspect of diagnosis and treatment planning. Further research exploring reasons for delaying seeking treatment following symptom onset in ethnic Indian women with PCOS is required to ensure timely diagnosis.

Long and irregular cycles are associated with an increased risk of developing T2D [44]. This is important because ethnic Indian women have a high genetic predisposition to develop T2D [19]. We found that 43% of women reported having a family history of T2D and 18% reported a family history of PCOS. Previous research suggests that PCOS is a disorder that runs in female relatives of the family associated with high IR and metabolic syndrome in male first-degree relatives [45,46,47]. Therefore, increased awareness and better screening tools for women with such a family history need to be carefully implemented.

In comparison with previous studies conducted on white women with PCOS about diagnosis satisfaction [21,22], women in our study reported greater dissatisfaction with the manner of diagnosis (49% vs. 34%) and emotional support (49% vs. 35%). We also found an increase in the percentage of women reporting receiving information regarding PCOS (40% vs. 87%) and emotional support (62% vs. 66%) [21]. While earlier studies [21,22] have investigated dissatisfaction related to the information given for lifestyle (42%), we carried out a comprehensive investigation related to the components of lifestyle—diet, exercise, and behavioural advice—and found that among the women who received the relevant information, most of them were dissatisfied. This suggests that although information provisions have improved over a period of time, it is still not meeting the information needs of women. High-quality information provided at the time of PCOS diagnosis can increase women’s ability to self-manage and prevent disease progression [48]. This is an opportunistic time to initiate engagement with evidence-based management strategies such as lifestyle change, which is the first-line treatment for PCOS, particularly for Indian women living overseas [49,50]. Thus, a greater understanding of the culturally-specific information and support needs are required in women with PCOS.

Despite women in our study reporting poor provision of lifestyle information by their healthcare provider, over two-thirds were following a specific diet and engaging in physical activity to manage their symptoms. Consistent with previous research, these findings suggest that ethnic Indian women are sourcing information to help them manage their PCOS from alternative avenues [23]. This is concerning because research suggests only a limited number of PCOS websites provide culturally appropriate, accurate, and reliable lifestyle management information in accordance with evidence-based guidelines [51,52]. Ensuring that the content provided to ethnic Indian women adequately addresses their concerns and key priorities, including issues surrounding weight, body image, and mental health, will help to improve satisfaction with health professional education and online resources.

As the majority of women also reported using ingestible TCIM to manage PCOS, these complementary approaches likely need to be better addressed by the concerned health professional. This should include the provision of evidence-based recommendations surrounding more traditional practices, namely yoga and Ayurveda, which are traditional medicines used frequently by Indian women [53]. The process for consumers to be involved in the prioritization of clinical questions in guideline updates and the need for different countries to ensure specific concerns are addressed in country-specific guideline adaptations should be encouraged in the future.

Ethnic and socio-cultural differences in the HRQoL in women with PCOS have been described in previous studies with varying results in each domain of the MPCOSQ [54,55]. In contrast to research in Caucasian women [31,56], the emotion domain of the MPCOSQ produced the second lowest score after weight, also showing moderate impairment on HRQoL (mean MPCOSQ score 3.0). Significantly lower scores in the emotion domain have been previously reported among Black women compared to white women in a study conducted in the USA [57]. We found a significant difference in the mean scores of weight in women living in India and outside India. This is significant considering ethnic Indian women with PCOS often present with lower BMI than Caucasians [15]. Health professionals should be careful not to disregard education surrounding weight management, including weight prevention, which is crucial in healthy weight women with PCOS, as it can lead to the prevention of weight gain and symptoms worsening. Understanding the impact of PCOS on QoL is key to delivering meaningful outcomes, ensuring that clinicians and women can work in partnership to address women’s priorities.

## 6. Strengths and Limitations

This novel study has several strengths. Through several online recruitment channels and paid advertisements on social media, we were able to reach a very large ethnic Indian sample in a community setting living worldwide. To the best of our knowledge, this is the largest dataset of ethnic Indian women with PCOS. As the survey was voluntary and anonymous, the results are likely to reflect the authentic views of women with PCOS in the community. We included women who were diagnosed with PCOS by a medical doctor. Findings from this study will inform an international initiative to improve diagnosis and education to better meet the ethnically diverse health needs of women living with PCOS and helping to tailor diagnostic assessments and educational resources [58]. This will further help to better characterize their health needs and to support culturally informed care globally.

Limitations of this study include the exclusion of women who were younger than 18 years of age and older than 55 years of age. PCOS continues to affect women in their adolescence and post-menopause; however, their experiences were not captured in this survey. This study also might have a risk of reporting bias, recall bias, and selection bias. As the cohort was relatively young, recall bias is less likely with the diagnosis occurring recently. Selection bias is possible due to all participants being recruited from websites and online groups, eliminating the potential to reach women with PCOS who are not active online. Our sample is also very highly educated, with 93% of women receiving a university education, and hence does not adequately capture the experiences of women from lower socioeconomic backgrounds. The survey was only available in English and posted on English websites, and may have excluded non-English speaking women. Additionally, we did not design the study as a case-control study, which limits the comparisons with non-Indian women. Findings related to the use of TCIM and the level of physical activity will be reported in future publications.

## 7. Conclusions

This study suggests that diagnostic delay still exists and that ethnic Indian women are waiting a considerable time before seeking healthcare advice about their PCOS symptoms, which is associated with dissatisfaction with diagnosis experience and poor emotional QoL. Indian women living overseas are most concerned about their weight. However, they are receiving inadequate information regarding lifestyle changes and are dissatisfied with the information given. Better information provision related to lifestyle and behavioural advice should be sought for women living outside India. Education at the time of diagnosis and focus on emotional well-being, as well as information on managing clinical manifestations that can affect body image, namely weight gain and hirsutism, are needed. While screening for and treatment of mood disorders is paramount in PCOS management of ethnic Indian women, health professionals should pay attention to possible family history and co-morbidities and ask about the concerns that matter most to the individual woman. High levels of engagement with TCIM also suggest that evidence-based recommendations for culturally relevant practices warrant further investigation. It is also important to consider sociocultural factors that may delay ethnic Indian women in seeking treatment, including the social stigma surrounding menstruation.

## 8. Future Recommendations

The survey results highlight the importance and need for a shift in clinical healthcare delivery to a multi-disciplinary, person-centred and family-focused, and comprehensive approach for this heterogeneous condition. Although international PCOS guidelines provide a great tool for healthcare professionals, a focus on local adaptation may help fill the social and cultural gaps to support culturally appropriate care. Women from ethnic Indian backgrounds experience an earlier onset of symptoms of PCOS, have a high rate of psychological comorbidity, and rate weight problems as their biggest concern, and these issues should be taken into consideration by their treating healthcare professionals. More research that can be used to better understand differences in not only PCOS presentations but also health-seeking behaviour in ethnic Indian women should be prioritized, as should research on psychological comorbidities. Incorporating consumer preferences, including engaging consumers in the co-design of educational resource development, will help to ensure management is individualized to suit the unique health goals of different populations. More research on the role of TCIM, particularly yoga and Ayurveda, in the management of PCOS is warranted.

## Figures and Tables

**Figure 1 ijerph-19-15850-f001:**
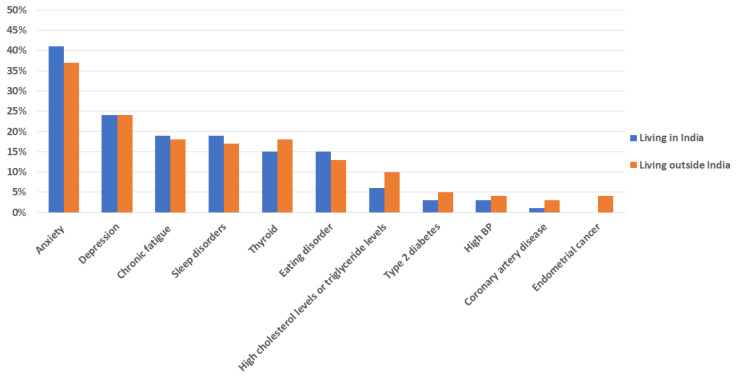
Co-morbidities reported by the participants (*n* = 1816).

**Figure 2 ijerph-19-15850-f002:**
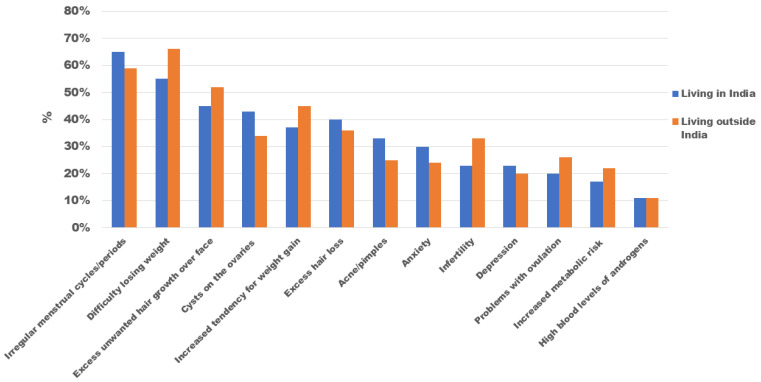
Key clinical features of PCOS of most importance to women (*n* = 2776).

**Figure 3 ijerph-19-15850-f003:**
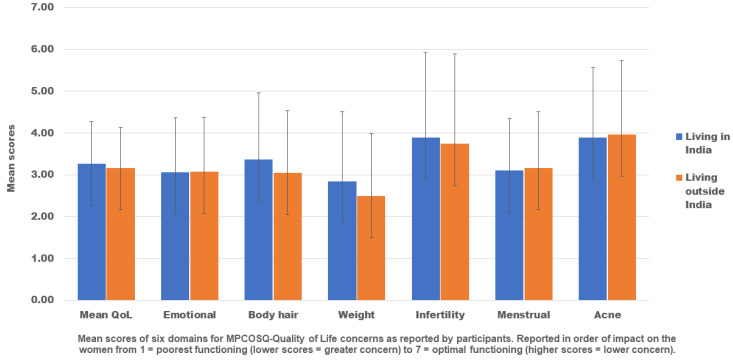
Health-related QoL (MPCOSQ scores) and country of residence (*n* = 2020).

**Table 1 ijerph-19-15850-t001:** Demographics of the participants.

	Total Responses, N (%)	Living in India, *n* (%)	Living Outside India, *n* (%)
**Age (years)**	**4409**	2046 (74)	734 (26)
18–24	1631 (37)	950 (46)	114 (16)
25–34	2367 (53)	963 (47)	489 (67)
≥35	411 (9.3)	133 (7)	131 (17)
**BMI (kg/m^2^)**	**4409**	-	-
<18.0	110 (3)	57 (3)	10 (2)
18.0–22.9	930 (21)	515 (25)	109 (15)
23.0–24.9	674 (15)	293 (14)	99 (13)
≥25.0	2695 (61)	1181 (58)	516 (70)
**Country of birth**	**2780**		
India	2577 (93)	1996 (98)	581 (79)
Other countries	203 (7)	50 (2)	153 (21)
Missing	1629 (37)	-	-
**Education**	**2780**		
Postgraduate degree	1213 (44)	767 (37)	375 (51)
Undergraduate degree	1375 (49)	1077 (53)	298 (41)
Below undergraduate degree	192 (7)	202 (10)	61 (8)
Missing	1629 (37)	-	-
**Occupation**	**2806**		
Employed	1521 (54)	1003 (49)	483 (66)
Studying	860 (31)	671 (33)	89 (12)
Unemployed	365 (13)	190 (9)	66 (9)
Home Duties	330 (12)	159 (8)	90 (12)
Other ^a^	110 (4)	21 (1)	5 (1)
Missing	1603 (36)	-	-
**Relationship status**	**2780**		
Married and living with a partner/In a relationship	1623 (58)	1058 (52)	565 (77)
Single	1113 (40)	960 (47)	153 (21)
Other ^b^	596 (21)	28 (1)	16 (2)
Missing	16,291 (37)	-	-
**History of pregnancy**	**2780**		
No	2185 (79)	1740 (85)	445 (61)
Yes	595 (21)	306 (15)	289 (39)
Missing	1629 (37)	-	-
If yes, ever needed treatment to fall pregnant	589		
• NO	307 (52)	151 (50)	156 (55)
• Yes	282 (48)	154 (50)	128 (45)
• Missing	6 (0.01)	-	-
If yes, the number of biological children	526		
• One	228 (43)	110 (41)	118 (45)
• Currently pregnant	160 (30)	73 (27)	87 (33)
• Two or more	73 (14)	30 (11)	43 (16)
• None	70 (13)	54 (20)	16 (6)
• Missing	69 (12)	-	-
**Family history of PCOS**	**3993**		
No	3274 (82)	1677 (82)	575 (79)
Yes	719 (18)	365 (18)	154 (21)
• Sister	513 (66)	249 (68)	116 (75)
• Mother	250 (32)	135 (37)	50 (32)
• Daughter	14 (2)	7 (2)	4 (3)
Missing	416 (9)	-	-
**Family history of type 2 diabetes**	**3963**		
No	2251 (57)	1161 (57)	386 (53)
Yes	1712 (43)	862 (43)	434 (47)
• Father	1290 (75)	632 (73)	231 (53)
• Mother	828 (48)	392 (45)	177 (41)
• Sister	40 (2)	11 (1)	14 (3)
• Brother	35 (2)	16 (2)	10 (2)
• Daughter or Son	5 (0.1)	3 (0)	2 (1)
Missing	446 (10)	-	-

^a^ Volunteer, retired, unable to work because of Polycystic Ovary Syndrome (PCOS). ^b^ Divorced, separated, widow. BMI = Body Mass Index.

**Table 2 ijerph-19-15850-t002:** Symptoms at onset and diagnosis.

	Overall Responses, Mean (SD)	Living in India, Mean (SD)	Living Outside India, Mean (SD)
Age when symptoms first appeared	19.0 (5.0)	18.5 (4.7)	19.6 (5.4)
Age when first visited a health professional	20.0 (5.0)	19.6 (4.6)	20.8 (5.2)
Age when diagnosed with PCOS	20.8 (4.8)	20.0 (4.4)	22.0 (5.4)
Sign/symptoms at onset, n (%)	3824		
Irregular menstrual cycles/periods	3156 (83)	1724 (85)	613 (84)
Cysts on the ovaries (in an ultrasound)	2263 (59)	1274 (63)	418 (57)
Excess unwanted hair growth on the face	1836 (48)	967 (47)	406 (55)
Increased tendency for weight gain	1820 (48)	986 (48)	376 (51)
Difficulty losing weight	1763 (46)	930 (46)	371 (51)
Acne/Pimples	1637 (43)	928 (46)	276 (38)
Excess hair loss	1443 (38)	821 (40)	233 (32)
Anxiety	757 (20)	421 (21)	136 (19)
Problems with ovulation	662 (17)	329 (16)	153 (21)
Depression	578 (15)	333 (16)	94 (12)
High blood levels of androgens/male hormones (e.g., testosterone)	511 (13)	279 (14)	121 (17)
Not able to fall pregnant (Infertility)	375 (10)	170 (8)	111 (15)
Other than above	189 (5)	117 (6)	34 (5)
Increased metabolic risk (e.g., fear of developing type 2 diabetes)	23 (1)	128 (6)	56 (8)
I do not remember	19 (0.1)	6 (0.3)	5 (0.7)
Missing	585 (13)	-	-
Diagnosed with PCOS by, n (%)	3839		
Gynecologists/obstetrician	3149 (82)	1746 (85)	543 (74)
General practitioner/family physician/family doctor	376 (9.8)	121 (6)	122 (17)
Endocrinologist	150 (3.9)	78 (4)	43 (6)
Dermatologist	82 (2.1)	47 (2)	8 (1)
Infertility specialist	77 (2.0)	42 (2)	17 (2)
Cardiologist	3 (0.001)	-	-
Psychiatrist	2 (0.001)	2 (0.1)	0 (0)
Missing	577 (13)	-	-

**Table 3 ijerph-19-15850-t003:** Information provision and satisfaction at the time of diagnosis.

	Overall Responses, (N = 3595)	Country of Residence (*n* = 2780)		*p*-Value *
	*n* (%)	Living in India, *n* (%)	Living Outside India, *n* (%)	
Information provision about				
PCOS				**0.039**
Information was not given	482 (13)	309 (15)	88 (12)	
Information was given	3113 (87)	1737 (85)	646 (88)	
Long-term complications				**0.042**
Information was not given	1240 (35)	749 (37)	238 (32)	
Information was given	2355 (65)	1297 (63)	496 (68)	
Treatment options				0.594
Information was not given	516 (14)	298 (15)	101 (14)	
Information was given	3079 (86)	1748 (85)	633 (86)	
Diet				0.655
Information was not given	871 (24)	507 (25)	188 (26)	
Information was given	2724 (76)	1539 (75)	456 (74)	
Exercise				**0.004**
Information was not given	737 (21)	409 (20)	184 (25)	
Information was given	2858 (79)	1637 (80)	550 (75)	
Behavioural advice to support diet and exercise				0.068
Information was not given	1071 (30)	625 (31)	251 (34)	
Information was given	2524 (70)	1421 (70)	483 (66)	
Emotional support after diagnosis				0.927
Information was not given	1232 (34)	737 (36)	263 (36)	
Information was given	2363 (66)	1309 (64)	471 (64)	
Overall satisfaction with the manner of diagnosis				
Dissatisfied	1067 (30)	618 (30)	229 (31)	0.314
Neutral	1304 (36)	763 (37)	251 (34)	
Satisfied	1224 (34)	665 (33)	254 (35)	
Satisfaction with the information given about	3113	2383		
PCOS				0.278
Dissatisfied	1038 (33)	591(34)	229 (36)	
Neutral	910 (29)	515 (30)	170 (26)	
Satisfied	1165 (38)	631 (36)	247 (38)	
Long-term complications	2355	1793		0.502
Dissatisfied	959 (41)	532 (41)	198 (40)	
Neutral	594 (25)	326 (25)	138 (28)	
Satisfied	802 (34)	439 (34)	160 (32)	
Treatment options	3079	2381		0.424
Dissatisfied	1533 (50)	875 (50)	334 (51)	
Neutral	691 (22)	400 (23)	131 (21)	
Satisfied	855 (28)	473 (73)	168 (26)	
Diet	2724	2085		**0.012**
Dissatisfied	1065 (24)	600 (39)	242 (44)	
Neutral	666 (15)	363 (24)	138 (25)	
Satisfied	993 (22)	576 (37)	546 (31)	
Exercise	2858	2187		**0.009**
Dissatisfied	939 (33)	518 (32)	202 (37)	
Neutral	725 (25)	416 (25)	152 (28)	
Satisfied	1194 (42)	703 (43)	196 (35)	
Behavioural advice to support diet and exercise	2524	1904		0.158
Dissatisfied	962 (38)	551 (39)	209 (43)	
Neutral	701 (28)	388 (27)	130 (27)	
Satisfied	861 (34)	482 (34)	144 (30)	
Emotional support after diagnosis	2363	1780		0.834
Dissatisfied	1239 (53)	716 (55)	256 (54)	
Neutral	530 (22)	273 (21)	104 (22)	
Satisfied	594 (22)	320 (24)	111 (24)	
Delay in seeking help after symptoms onset	3276	2325		0.263
Less than 1 year	1707 (52)	897 (52)	292 (49)	
1 year and above	1569 (48)	834 (48)	302 (51)	
Delay in diagnosis	3195	2320		0.296
Less than 1 year	2340 (73)	1249 (74)	443 (71)	
1 year and above	855 (27)	450 (26)	178 (29)	

* chi-square test; significant *p*-values are marked in bold.

**Table 4 ijerph-19-15850-t004:** Association between the place of residency, delay in diagnosis and seeking help with diagnosis experiences.

	Country of Residency: India ^a^/Outside India	Delay in Diagnosis: Less Than 1 Year ^a^/1 Year or More	Delay in Seeking Help: Less Than 1 Year ^a^/1 Year or More
	OR (95% CI), *p*-Value	OR (95% CI), *p*-Value	OR (95% CI), *p*-Value
Overall satisfaction with the manner of diagnosis	1.01 (1.0 to 1.2), 0.722	0.69 (0.5 to 0.8), <0.001 *	0.91 (0.7 to 1.0), 0.270
Information provided about ^b^			
PCOS	1.40 (1.0 to 1.8), 0.014 *	0.62 (0.4 to 0.8), <0.001 *	0.97 (0.7 to 1.2), 0.836
Long-term complications	1.23 (1.0 to 1.4), 0.030 *	0.81 (0.6 to 0.9), 0.031 *	1.0 (0.8 to 1.1), 0.938
Treatment options	1.18 (0.9 to 1.5), 0.197	0.79 (0.6 to 1.0), 0.082	0.95 (0.7 to 1.2), 0.693
Diet	1.04 (0.8 to 1.2), 0.711	0.91 (0.7 to 1.1), 0.399	0.96 (0.7 to 1.4), 0.670
Exercise	0.83 (0.6 to 1.0), 0.106	0.90 (0.7 to 1.1), 0.390	0.77 (0.6 to 0.9), 0.011 *
Behavioural advice to improve diet and exercise	0.91 (0.9 to 1.1), 0.368	1.00 (0.8 to 1.2), 0.986	0.88 (0.7 to 1.0), 0.183
Emotional support after diagnosis	1.02 (0.8 to 1.2), 0.775	1.04 (0.8 to 0.2), 0.682	0.83 (0.7 to 0.9), 0.037 *
Level of satisfaction with the information provided about ^c^			
PCOS	0.97 (0.8 to 1.1), 0.722	0.68 (0.5 to 0.8), <0.001 *	0.86 (0.7 to 1.0), 0.069
Long-term complications	0.96 (0.7 to 1.1), 0.726	0.70 (0.5 to 0.8), <0.001 *	0.94 (0.7 to 1.1), 0.577
Treatment options	0.84 (0.6 to 1.0), 0.065	0.73 (0.6 to 0.8), 0.002 *	0.71 (0.6 to 0.8), <0.001 *
Diet	0.74 (0.6 to 0.8), 0.002 *	0.74 (0.6 to 0.9), 0.004 *	0.80 (0.6 to 0.9), 0.013 *
Exercise	0.74 (0.6 to 0.9), 0.002 *	0.68 (0.5 to 0.8), <0.001 *	0.86 (0.7 to 1.0), 0.101
Behavioural advice to improve diet and exercise	0.74 (0.6 to 0.9), 0.004 *	0.72 (0.5 to 0.8), 0.002 *	0.69 (0.5 to 0.8), <0.001 *
Emotional support after diagnosis	0.87 (0.7 to 1.0), 0.212	0.78 (0.6 to 0.9), 0.034 *	0.62 (0.5 to 0.7), <0.001 *

Multivariable analysis adjusted for country of residence, age, BMI, occupation, education, and relationship status. ^a^ = Reference group, ^b^ = Not given ^a^/given, ^c^ = Dissatisfied/neutral/satisfied ^a^; * Significant *p*-value; PCOS = Polycystic Ovary Syndrome; OR = Odds ratio; CI = Confidence interval.

**Table 5 ijerph-19-15850-t005:** Health professionals consulted and treatment methods used to manage PCOS.

	Total Responses, n (%)	Living in India	Living Outside India
The number of health professionals consulted	3117		
One	1444 (46)	958 (48)	287 (40)
Two	1368 (44)	877 (44)	338 (47)
Three or more	305 (10)	180 (9)	92 (13)
List of health professionals consulted	3117		
Gynecologist/obstetrician	2809 (90)	1855 (91)	611 (83)
General practitioner/family physician/family doctor	825 (26)	421 (21)	313 (43)
Allied health professional (e.g., dietician, exercise physiologist)	621 (20)	399 (20)	148 (20)
Endocrinologist	518 (17)	306 (15)	157 (21)
Dermatologist	505 (16)	356 (17)	89 (12)
Infertility specialist	350 (11)	201 (10)	118 (16)
Psychiatrist	122 (4)	78 (4)	32 (4)
Cardiologist	14 (0.1)	7 (0.3)	4 (0.5)
Other than above	225 (7)	168 (8)	37 (5)
Never seen a medical doctor or allied health practitioner for PCOS treatment	58 (2)	31 (2)	17 (2)
Missing	1292 (29)	-	-
List of conventional treatments used	2547		
Combined oral contraceptive pills (estrogen + progestin)	1613 (63)	1052 (64)	389 (63)
Metformin (insulin-sensitizing medicines)	1048 (41)	593 (36)	332 (54)
Anti-androgen drugs (to correct male-hormone levels)	637 (25)	436 (27)	131 (21)
Ovulation induction to fall pregnant (e.g., Letrozole, Clomid, gonadotropins)	359 (14)	193 (12)	133 (22)
Anti-obesity drugs	174 (7)	115 (7)	41 (7)
Intrauterine insemination (IUI)	131 (5)	83 (5)	34 (6)
Laparoscopic surgery (ovarian drilling)	123 (5)	63 (4)	52 (8)
In-vitro fertilization (IVF) or intracytoplasmic sperm injection (ICSI)	99 (4)	41 (3)	48 (8)
Intrauterine device (IUD) (e.g., Mirena or Depo Provera)	39 (2)	11 (1)	24 (4)
Bariatric surgery	15 (1)	2 (0.1)	8 (1.3)
Other than above	463 (18)	337 (21)	71 (12)
Missing	1862 (42)	-	-
Diets used to manage PCOS in the past five years	2921		
Yes	1978 (68)	1328 (65)	558 (76)
No	943 (32)	718 (35)	176 (24)
Missing	1448 (33)	-	-
Use of exercise	2921		
Yes	2241 (77)	1572 (77)	540 (74)
None of the given forms	680 (23)	474 (23)	194 (26)
Missing	1488 (34)	-	-
Use of ingestible TCIM	3098		
Yes	1942 (63)	1345 (66)	399 (54)
No	1156 (37)	701 (34)	335 (46)
Missing	1279 (29)	-	-
Use of yoga	3027		
Yes	1775 (59)	1238 (61)	367 (50)
No	1252 (41)	808 (39)	367 (50)
Missing	1382 (31)	-	-

## Data Availability

The datasets used and/or analysed during the current study are available from the corresponding author upon reasonable request.

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
