# Peer review of "A Global Survey of Ethnic Indian Women Living with Polycystic Ovary Syndrome: Co-Morbidities, Concerns, Diagnosis Experiences, Quality of Life, and Use of Treatment Methods"

_ijerph, 2022, doi:10.3390/ijerph192315850_

Round 1

Reviewer 1 Report

This is an excellent and well-written global online survey of ethnic Indian women of reproductive age living with PCOS.  The group of authors should be commended for such work.  I have no major suggestions for improvement. 

I do suggest the authors add some commentary to their manuscript on the findings of the following publication on PCOS from 2021 -- PMID: 33658043

Reviewer 2 Report

Dear Editors and authors,

I am grateful to revise this paper.

The objective of this article is a survey regarding the clinical impact and awareness of PCOS by Indian women.

In my opinion, this paper could be suitable for publication after few revisions.

In fact, it could be improved as follows:

·      Introduction: according to many studies, the ethnicity could influence the development of PCOS. The authors could cite Belenkaia LV, Lazareva LM, Walker W, Lizneva DV, Suturina LV. Criteria, phenotypes and prevalence of polycystic ovary syn[1]drome. Minerva Ginecol. 2019;71(3):211–23 in order to underline that PCOS is usually more frequent in Mexican American and Caucasian subjects rather than South Chinese and Asian women. Anyway, the prevalence of this syndrome may vary according to several aspects, that is to say ethnicity as well as applied diagnostic criteria.

Additionally, I suggest pointing out the putative mechanisms underlying the augmented incidence of diabetes type 2 reported in Indian women compared to Caucasian people.

·      Discussion: in my opinion, the authors could improve the quality of their manuscript discussing the importance of family history in clinical practice. In particular, they could report and discuss data about the recurrence of hyperandrogenic trait within the same family to explain, at least in part, the results of their survey regarding the percentages of sisters and mothers of PCOS patients affected by the syndrome. (See Kahsar-Miller MD, Nixon C, Boots LR, Go RC, Azziz R. Prevalence of polycystic ovary syndrome (PCOS) in first-degree relatives of patients with PCOS. Fertil Steril. 2001;75(1):53–8)

As for the sentence “Most women experienced signs and symptoms of PCOS between the ages of 13-20 years and received a PCOS diagnosis between the ages of 17-24 years”, the authors should consider the possible concerns about an actual diagnosis of PCOS in adolescence. In fact, even if this is a survey taking into account self-reported diagnosis, they should mention and discuss some recent studies highlighting this intriguing but challenging aspect. (See and cite  Capozzi A, Scambia G, Lello S. Polycystic ovary syndrome (PCOS) and adolescence: How can we manage it? Eur J Obstet Gynecol Reprod Biol. 2020 Jul;250:235-240. doi: 10.1016/j.ejogrb.2020.04.024. Epub 2020 Apr 15. PMID: 32497923.;

Witchel SF, Burghard AC, Tao RH, Oberfield SE. The diagnosis and treatment of PCOS in adolescents: an update. Curr Opin Pediatr. 2019 Aug;31(4):562-569. doi: 10.1097/MOP.0000000000000778. PMID: 31299022)

Sincerely

Reviewer 3 Report

The manuscript of Vibhuti Rao and collaborators aims to perform an analysis of an online survey of ethnic Indian women of reproductive age living with Polycystic ovary syndrome. I believe this manuscript requires much more planning in the analysis. The following revisions must be instituted in order to bring the manuscript to publication quality: In its current form, the manuscript reads like a collection of facts, rather than a detailed analysis geared towards explaining what are the concerns, diagnosis experiences, quality of life, and use of treatment methods. In other words, the manuscript is unfocused and results are modest and poorly presented. Graph presentation requires further improvements and detailed analysis (for example: figure 3 requires SD). Therefore, while the subject matter is important and significant, the manuscript lacks clarity and may not add significantly to the existing literature on this subject as the major conclusion is similar to other publications.

Round 2

Reviewer 3 Report

Authors have improved the manuscript, but unfortunately, it has still very weaknesses and does not add significantly to the existing literature on this subject
